# Shear Wave Elastography of the Sciatic Nerve and Its Relationship with Posterior Chain Flexibility in Healthy Participants: An Observational Study

**DOI:** 10.3390/s25092885

**Published:** 2025-05-02

**Authors:** Charles Cotteret, Jaime Almazán-Polo, Ángel González-de-la-Flor

**Affiliations:** Department of Physiotherapy, Faculty of Medicine, Health and Sports, European University of Madrid, 28670 Villaviciosa de Odón, Spain; charles.cotteret@universidadeuropea.es (C.C.); angel.gonzalez@universidadeuropea.es (Á.G.-d.-l.-F.)

**Keywords:** ultrasound, shear wave elastography, posterior chain, flexibility, sciatic nerve

## Abstract

Introduction: Shear wave elastography (SWE) has been widely used to assess the mechanical properties of peripheral nerves, including the sciatic nerve. However, the relationship between sciatic nerve stiffness and posterior chain flexibility remains unclear. Therefore, we aimed to examine differences in sciatic nerve stiffness and shear wave speed (SWS) based on limb dominance and hamstring flexibility, and to explore their association with posterior chain mobility assessed through AKE and ASLR tests in healthy individuals. Methods: An observational study was conducted on 25 healthy, physically active participants (49 lower limbs). Sciatic nerve stiffness was measured using SWE at a standardized location in the posterior thigh. Posterior chain flexibility was assessed using the Active Knee Extension (AKE) and Active Straight Leg Raise (ASLR) tests. Participants were categorized based on hamstring flexibility, and comparisons were made between dominant and non-dominant limbs. Results: Participants with limited hamstring flexibility exhibited significantly higher AKE and ASLR values (*p* < 0.001) and showed an increased stiffness and SWS towards greater sciatic nerve (*p* = 0.05), although correlations between SWE values and flexibility tests were not significant. No significant differences were found between dominant and non-dominant limbs in AKE (*p* = 0.28), ASLR (*p* = 0.47), SWE (*p* = 0.38), or SWS (*p* = 0.34) values. Conclusions: Although no significant correlations were found between SWE parameters and flexibility tests, individuals with limited posterior chain mobility exhibited higher sciatic nerve stiffness in healthy participants.

## 1. Introduction

Muscle flexibility, defined as the ability of muscle tissue to elongate, is a key factor in physical fitness and the prevention of musculoskeletal dysfunctions [1]. Specifically, hamstring tightness is associated with a reduced range of motion (ROM) in the hip and/or knee, which can lead to biomechanical alterations, increase the risk of injuries, and negatively impact daily activities [2,3]. Among university students, hamstring stiffness is a common issue, with a high prevalence (68%) reported in young adults aged 18 to 25 years, where sedentary behavior and prolonged sitting are key contributing factors [4].

The assessment of posterior chain flexibility is conducted using various methods, including the straight leg raise (SLR) and knee extension angle measurements in both active (AKE) and passive (PKE) conditions [1,5]. The AKE and PKE tests provide a more specific evaluation of hamstring flexibility by stabilizing the hip joint. The AKE, one of the most widely used measures for assessing hamstring tightness, considers knee extension angle as an indirect indicator of flexibility [6]. Beyond flexibility, hamstring shortening not only affects joint mobility but also alters posture, inducing posterior pelvic tilt and reducing lumbar lordosis [7].

Elastography is an ultrasound-based technique used to assess the mechanical properties of tissues, particularly their stiffness. Two main elastography techniques exist: strain elastography (SE) and shear wave elastography (SWE) [8,9]. SE measures tissue displacement in response to mechanical compression, whereas SWE analyzes the propagation of shear wave speed (SWS) induced within the tissue, allowing for a quantitative estimation of stiffness, expressed in meters per second (m/s) or kilopascals (kPa) [8,10,11,12].

Previous studies have shown that these postural changes may be associated with increased stiffness in neural structures such as the median, tibial, and sciatic nerves [7,13]. In this context, SWE has become an increasingly popular imaging modality for evaluating peripheral nerves, providing valuable insights into their microstructural characteristics. As a form of sensor-captured imaging technology, SWE enables the non-invasive quantification of tissue biomechanical properties through the acquisition of real-time mechanical wave propagation data. This highlights its utility as a sensor-integrated approach within ultrasound imaging, with broad applications in musculoskeletal and neurological assessments [14].

SWE has demonstrated good reliability in the assessment of peripheral nerve stiffness. Specifically, for the sciatic nerve, previous studies have reported excellent reliability when using standardized protocols [15,16]. SWE has also been validated in evaluating the stiffness of the median, tibial, and sciatic nerves under both passive and active conditions, as well as in differentiating between affected and unaffected limbs in neuropathic populations [15,16,17,18]. These findings support the utility of SWE for detecting mechanical changes in neural tissue. Nonetheless, it should be noted that reliability may be influenced by factors such as operator experience and anatomical variability [19].

Although shear wave elastography (SWE) has demonstrated good reliability, several technical and anatomical factors must be considered when interpreting its results. These include probe positioning, compression levels, measurement site, anatomical variability, and the degree of neural tension applied during assessment [8,20]. Notably, previous studies have shown that shear wave speed (SWS) tends to increase under specific conditions, highlighting the sensitivity of SWE to both positioning and biomechanical context [13,14,20].

Similar phenomena have been observed in the sciatic nerve, where an increase in SWS was detected following tension application through passive ankle dorsiflexion combined with passive knee extension [8,17]. However, most previous SWE studies evaluating the mechanical properties of peripheral nerves have used passive movements to investigate the relationship between function and stiffness. In clinical and research settings, these findings may not fully reflect real-world scenarios, where exercises are typically performed actively [21].

In recent years, several studies have examined the effect of postural changes on sciatic nerve stiffness using SWE [13]. However, there is a lack of evidence regarding the potential relationship between the biomechanical properties of neural tissue and functional variables such as the AKE and ASLR tests. Moreover, the effect of hamstring shortening on stiffness and SWS has not yet been studied.

Therefore, we aimed to determine whether differences exist in sciatic nerve stiffness and SWS, assessed via SWE, and posterior chain mobility, measured through the AKE and ASLR tests, between dominant and non-dominant limbs, as well as between individuals with limited and adequate hamstring flexibility in healthy individuals. Lastly, we aimed to explore the association between SWE-derived parameters and AKE and ASLR range of motion.

## 2. Methods

### 2.1. Study Design

A cross-sectional observational study was conducted, adhering to the Strengthening the Reporting of Observational Studies in Epidemiology (STROBE) guidelines and checklist for cross-sectional studies [22]. The study protocol received approval from the local Research Committee of the European University of Madrid (CI internal code: 2025-399), and all ethical considerations were in accordance with the recommendations outlined in the Declaration of Helsinki.

### 2.2. Sample Size Calculation

Based on previous studies [15,23], an a priori sample size estimation was conducted using G*Power (version 3.1.9.7), considering a two-tailed t-test for the difference between two independent means (limited vs. adequate flexibility). With an expected effect size of *d* = 0.85, a significance level of α = 0.05, and a statistical power of 0.80, the required total sample size was calculated to be 46 legs. Therefore, our final sample of 49 legs meets the minimum requirement to detect statistically meaningful differences between groups.

### 2.3. Participants

Participants were recruited and screened for potential eligibility between January and March 2025. The study was conducted with a cohort of healthy, physically active students from the European University of Madrid. Participants were recruited through a combination of flyer distribution, poster placements, targeted advertisements within the university premises, and email campaign outreach. Eligibility criteria included both male and female individuals aged 18 to 30 who maintained a regular training regimen of at least two days per week. Exclusion criteria encompassed a history of musculoskeletal conditions affecting the lower limbs or lumbopelvic region within the past five years, as well as the presence of neuromuscular, rheumatic, cardiovascular, or neurological disorders. Additionally, individuals with a history of prior surgical interventions or fractures in the lower extremities or an inability to obtain SWE measurements of the sciatic nerve due to poor image quality were not eligible for participation.

### 2.4. Descriptive Variables

The participant’s gender (male or female), age (in years), height (in centimeters), weight (in kilograms), dominance limb (right or left) [24], body mass index (BMI), which is determined in kg/cm^2^ using Quetelet’s index [25], depth of the sciatic nerve (in centimeters), and hamstring flexibility (in degrees) are all included in the sociodemographic data. Additionally, data on the distribution of the dominant limb (right or left limb) were also gathered. Participants were categorized into the hamstring flexibility restriction subgroup based on the AKE test. Specifically, those who presented a test value below 33° for men and 23.4° for women were assigned to the “Restriction” subgroup, while those with values above these thresholds were classified into the “Adequate” subgroup [1].

### 2.5. Outcome Measures

All participants underwent a standardized familiarization session before data collection. This session ensured that each individual was acquainted with the testing procedures, particularly the AKE and ASLR tests, to mitigate learning effects and improve measurement consistency. Additionally, the order of test administration was randomized to prevent systematic errors related to fatigue or adaptation. Randomization was conducted using computer-generated sequences to allocate the sequence of flexibility tests and SWE assessments, ensuring that measurement bias was minimized.

#### 2.5.1. SWE Assessment for Sciatic Nerve

A unique expert sonographer with more than five years of experience in musculoskeletal ultrasound performed all examinations in the research laboratory of the European University of Madrid. The LOGIQ P9 XDCLEAR R4.5 ultrasound system (General Electric, Boston, MA, USA) was employed in conjunction with the SWE P9R3 elastography system, using the L3-12RS ultrasound probe, which operates within a frequency range of 3 to 12 MHz.

The sciatic nerve was first identified in the transverse plane by scanning the posterior thigh in B-mode (Figure 1C), approximately two finger widths below the gluteal fold (Figure 1A,B). The transducer was then oriented longitudinally until both the superficial and deep epineurium of the nerve were observed (Figure 1D). The detailed protocol for the identification and measurements of the sciatic nerve was as follows: (i) localization of the intermuscular septum between the long head of the biceps femoris (BFLH) and the semitendinosus (ST) using the short axis of the probe, with sweeping scans performed along the sciatic nerve; (ii) subsequently, the probe axis was adjusted to visualize the sciatic nerve in the long axis and assess its fascicular pattern (Appendix A). This specific location was chosen for stiffness measurement because, at this level, the sciatic nerve has not yet given off numerous articular (hip/knee) and muscular branches [19].

In this position, SWE was initiated (Figure 2), using the default preset with a frequency range of 100–500 Hz, a singular value decomposition (SVD) of 3.2 points, and a gain setting of 75 dBs. The SWE system enabled the selection of a 2 mm^2^ (white rectangular Q-box 1 mm × 2 mm, Figure 2) region of interest (ROI) within the elastography window, providing average values for the Young Modulus (kPa) and shear wave speed (SWS, m/s). The size of the Q-box was kept constant for all subjects in order to standardize the measurements. The final measurement was determined as the average of three consecutive images to assess intra-class reliability [26,27].

Minimal and consistent pressure was applied during all measurements to prevent excessive tissue compression [28], and SWE clip videos of approximately 30 s were recorded. Three elastographic images were acquired consecutively without removing or repositioning the transducer, and the average of these was used for analysis. Both SWS and Young’s modulus were analyzed in this study. Both Young’s modulus and SWS were directly obtained from the elastography system.

#### 2.5.2. Active Knee Extension Test

For the Active Knee Extension (AKE) Test, participants lay supine on a treatment table with their hip and knee flexed at 90 degrees. They were instructed to actively extend the knee while keeping the foot relaxed in plantar flexion, ensuring that the femur remained stationary by monitoring its position with their hand. The participants were directed to extend their leg as far as possible while keeping their foot relaxed, maintaining the extended position for 5 s. To ensure consistency, each participant performed a familiarization trial before a final test repetition. At the end of a 5 s hold in the extended position, knee extension was measured using a standard goniometer (Physiomed, Manchester, UK), with its center aligned to the lateral joint line and arms positioned along the femur and fibula markings. The measurement was taken within 2 s of reaching the end range of motion to standardize the duration of the static stretch. The AKE test has demonstrated excellent reliability for assessing hamstring flexibility [1,29]. Three consecutive angle measurements were taken on each leg using a goniometer, and the average of the three values was used for analysis.

#### 2.5.3. Active Straight Leg Raise

For the Active Straight Leg Raise (ASLR) test, participants were positioned on a standard examination table in a supine posture and instructed to stay relaxed throughout the procedure. The tested leg was flexed with the knee fully extended and the foot relaxed, while the examiner maintained the opposite leg in a fully extended, neutrally rotated position. The test was terminated when significant resistance was encountered or when pelvic rotation was observed. A goniometer was placed over the greater trochanter, with one arm aligned to the lateral femoral condyle and the other extending parallel to the table toward the mid-axillary line. The ASLR test has demonstrated high reliability for assessing posterior chain flexibility [30]. For each leg, three consecutive goniometric measurements were performed, and the average value was used in the analysis.

### 2.6. Statistical Analysis

Data analysis was conducted using the Statistical Package for the Social Sciences, SPSS (version 29.0; IBM Corp., Armonk, NY, USA), with a significance level set at *p*-value < 0.05. Normality was assessed using the Shapiro–Wilk test. For normally distributed variables, data were reported as mean ± standard deviation (SD), while non-normally distributed variables were presented as median and interquartile range (IQR).

Comparisons between groups were carried out to evaluate potential differences based on limb dominance (dominant vs. non-dominant) and hamstring flexibility (limited vs. adequate). For each comparison, independent samples t-tests were used when the data followed a normal distribution, and Mann–Whitney U tests were applied for non-normally distributed variables. Cohen’s *d* effect sizes were calculated to quantify the magnitude of differences between groups. Thresholds for standardized difference statistics were defined as follows: trivial (<0.20), small (0.20–0.59), moderate (0.60–1.19), large (1.20–1.99), and very large (>2.0) [31].

Correlations between AKE, ASLR, and SWE parameters (kPa and m/s) were assessed using Pearson or Spearman correlation coefficients, depending on data normality. AKE and ASLR values correspond to the goniometric angles measured during each test. The correlation analysis was conducted using data from all included limbs (both dominant and non-dominant). The strength of associations was classified as negligible (0.0 to 0.30), low (0.30 to 0.50), moderate (0.50 to 0.70), high (0.70 to 0.90), and very high (0.90 to 1.00) [32].

## 3. Results

A total of 25 participants (49 limbs) were included in the analysis. One leg was excluded from the analysis due to abnormal values detected in the SWE measurements. Descriptive characteristics of the sample, including anthropometric data, sex, and limb dominance, are presented in Table 1.

### 3.1. Comparison Between Dominant and Non-Dominant Leg

The analysis revealed no significant differences between limbs for AKE, ASLR, or SWE parameters (*p* > 0.05), with trivial effect sizes for these comparisons (Table 2).

### 3.2. Comparison Between Limited and Adequate Hamstring Stiffness Groups

Significant differences were observed between participants with limited and adequate hamstring stiffness in AKE and ASLR measures, with large to very large effect sizes (*d* = 1.61 to 2.12). Participants in the limited group exhibited higher values compared to the adequate flexibility group. SWE stiffness and speed also showed significant differences with higher values in the limited group (*p* = 0.05), with a moderate effect size (*d* = 0.60 to 0.61) (Table 3).

### 3.3. Correlation Analysis

Figure 3 and Table 4 present the correlation between sciatic nerve SWE and AKE and ASLR tests. Both Young’s modulus (kPa) and shear wave speed (SWS, m/s) were analyzed in relation to the goniometric angles obtained from the flexibility assessments.

The analysis showed a non-significant correlation between AKE and both SWE stiffness (r = 0.245, *p* = 0.089; Figure 3A) and SWS (r = 0.251, *p* = 0.082; Figure 3B). Similarly, no significant associations were found between ASLR and SWE stiffness (r = 0.119, *p* = 0.416; Figure 3C) or SWS (r = 0.110, *p* = 0.453; Figure 3D).

## 4. Discussion

### 4.1. General Results

The main findings of this study were that participants with limited hamstring flexibility exhibited significantly higher values in the AKE and ASLR tests (*p* < 0.001). Additionally, significant differences were found, indicating increased sciatic nerve stiffness and speed in these individuals (both, *p* = 0.05). Interestingly, no significant correlations were found between flexibility and SWE parameters, suggesting that other factors may influence these measurements. Furthermore, no significant differences were found in the posterior chain ROM or sciatic nerve stiffness based on limb dominance. The present study found no significant differences in the posterior chain range of motion or sciatic nerve stiffness between dominant and non-dominant limbs. These findings align with previous research suggesting that, in healthy individuals, limb dominance does not strongly influence passive or active flexibility nor neural mechanical properties [1]. The absence of significant differences (*p* > 0.05) and the trivial effect sizes observed for all variables reinforce the notion that functional or structural asymmetries may not be present in untrained populations. This supports the idea that clinical assessments or interventions aimed at improving posterior chain flexibility can be generalized across limbs, regardless of dominance.

To the best of our knowledge, this is the first study to investigate the relationship between sciatic nerve properties assessed through elastography and active posterior chain flexibility. While prior research has investigated nerve excursion and displacement during active or passive movements [21], they did not specifically analyze with the AKE and ASLR test, and correlations between SWE parameters and functional flexibility outcomes.

### 4.2. SWE of Sciatic Nerve Stiffness and SWS

In the AKE and ASLR measures, the participants in the limited group exhibited higher values compared to the adequate flexibility group. Likewise, SWE stiffness and speed were also significantly higher in the limited hamstring flexibility group. Although these are healthy subjects with limited hamstring flexibility, these findings align with previous research suggesting tissue changes associated with pathology versus normal tissues [8]. People with chronic low back-related leg pain have interlimb differences in sciatic nerve stiffness, with an increase in sciatic nerve stiffness in the affected leg compared to the non-affected side [8]. The observed changes may be associated with alterations in nerve mechanical properties, as healthy nerves exhibit greater stretch capacity due to their viscoelastic properties [17]. This trend may reflect a neural adaptation mechanism, where reduced flexibility leads to increased baseline nerve stiffness, potentially due to altered viscoelastic properties or reduced nerve excursion capacity. Alternatively, secondary stiffness could arise from surrounding soft tissue restrictions that limit nerve mobility.

In a recent systematic review, the normative values found for the mean SWS in the sciatic nerve were 6.70 ± 1.26 m/s in healthy controls and 7.51 ± 1.73 m/s in individuals experiencing leg pain [9].

In healthy subjects, no significant differences were found in the sciatic nerve SWS, with values of 6.93 m/s on the left side and 7.02 m/s on the right [8]. Once again, these results demonstrate greater stiffness, which results can be explained by the location of the sciatic nerve, as their measurements were taken 10 cm below the gluteal fold, more distally compared to our study. In a recent study by Cornelson et al., measurements were taken at four points, the first near the ischial tuberosity and the last just before the division of the sciatic nerve into the tibial nerve. The results showed that SWS values increased significantly from proximal to distal, with significant differences [33]. In addition to the measurement point, the equipment and software used in our study were different from those used by previous research [8]. The substantially lower SWS values observed in our study (3.1–3.4 m/s) compared to values reported in previous studies (6–10 m/s) can be attributed to several methodological differences [33]. These include the use of a proximal measurement site, supine positioning with reduced neural tension (neutral hip and plantar-flexed ankle), and minimal probe compression. For example, Zardi et al. found a stiffness range of 12.78–15.63 kPa for the sciatic nerve at the proximal level [19]. A possible explanation for this gradual increase could be the proximity of the sciatic nerve to the distal femur. Kantarci et al. found an increase in median nerve stiffness as it approached the carpal tunnel, attributing this finding to the “bone proximity” artifact [18,34]. Another key consideration is the ultrastructure of the sciatic nerve, which exhibits heterogeneous mechanical properties along its course, with an elliptical cross-sectional profile in its proximal portion. Furthermore, in highly mobile regions such as the hip or knee, nerves are subjected to increased mechanical strain, potentially leading to distinct shear wave velocities [35]. All these factors are known to decrease measured nerve stiffness.

Tang et al. reported similar findings, observing sex-based differences in stiffness; however, no statistically significant correlation was found. One potential explanation for this discrepancy involves the influence of subcutaneous cellular tissue, as well as methodological limitations in comparing the same nerve across distinct anatomical locations [36].

In addition to changes in nerve stiffness due to alterations in its properties under pathological conditions, it is well-established that such stiffness is influenced by neural tension, which varies according to limb position and joint angles. Andrade et al. reported that the sciatic nerve exhibited its greatest stiffness (10.4 ± 2.4 m/s) during knee extension (180°) with full ankle dorsiflexion, a condition that maximizes neural tension [17]. The decrease in sciatic nerve stiffness was significantly correlated with changes in the maximum range of motion in dorsiflexion (SWS = 7 m/s with 34° ankle dorsiflexion) [23]. Furthermore, Neto et al. found similar results, with SWS values ranging from 7 to 10 m/s depending on the percentage of dorsiflexion (0–100%) [15]. In a later study, SWS data were 8.67 m/s in healthy subjects at 80% of the ankle ROM [8].

Additionally, in relation to the previously mentioned location of the probe, the posture of the subjects could explain the differences found in the SWS in our study. The velocities found ranged from 3.1 to 3.4 m/s, far lower than the 7–10 m/s reported in the current literature [15]. Since subjects were in supine position, they were in a neutral hip position with nearly maximal ankle plantar flexion, which reduces the neural tension of the sciatic nerve, and therefore its stiffness.

Despite significant differences observed between individuals with limited and adequate flexibility in all variables, no significant correlations were found between SWE parameters and the AKE or ASLR tests. All correlation coefficients were in the negligible range (r < 0.30), suggesting that greater nerve stiffness does not directly translate into reduced posterior chain flexibility, at least in healthy participants. Several studies focused on nerve excursion during passive or active mobilizations and on interventions that influence nerve stiffness, without evaluating possible correlations between the mechanical properties of the structure and musculoskeletal function [7,14,15,18,21,37].

Ciuffreda et al. reported that the actual neural tension cannot be isolated from the effects of other forces acting upon the nerve, and therefore, surrounding tissues may influence the resultant nerve stiffness [13]. The lack of correlation between SWE parameters and flexibility tests may be explained by the multifactorial nature of posterior chain flexibility, which involves not only neural but also muscular, fascial, and articular structures [3,23,38,39]. In addition, the absence of subgroup analyses by sex or BMI was due to the small sample size, limiting the statistical power to detect potential hidden relationships. Future studies with larger samples should explore these subgroup comparisons.

### 4.3. Hamstrings Flexibility

Regarding the relationship between limb dominance and posterior chain flexibility, the literature presents varied findings. Consistent with our results, Yildirim et al. reported no significant differences in mean AKE angles between the dominant and non-dominant sides in a sample of 123 healthy university students. However, their reported values were lower than ours (indicating greater flexibility), with normative values of 17.8° ± 9.1° in men and 13.4° ± 6° in women. These discrepancies may be attributed to sample size and composition, as our sample had a lower proportion of women (33%), who are known to have greater posterior chain flexibility [1]. Similarly, Lim and Park found no differences in AKE and ASLR test results based on limb dominance. However, they reported greater posterior chain mobility compared to our study, with angles of 15° in AKE and 17° in ASLR. These differences may be explained by population-specific factors, as their study was conducted on the Korean population. Additionally, one of their inclusion criteria required participants to achieve a minimum of 33 cm in the sit-and-reach test. Consequently, individuals with limited flexibility were excluded from their study [39].

Vaquero-Cristóbal et al. observed significant differences in hamstring extensibility between ballet dancers and non-dancers, as expected. Ballet dancers had the greatest hamstring extensibility, with higher ASLR values and no significant differences between limbs in both groups. Their findings, with an average ASLR of 17°, were similar to the 21° reported in our results. Again, the higher proportion of women in their study could explain their slightly lower mean values [40].

Corkery et al. found greater hamstring flexibility in women, but no significant differences between right and left sides in a sample similar to ours, consisting of university students of the same age and BMI. In addition to gender, age, and BMI, hamstring flexibility can be influenced by other modifiable factors, such as physical activity level [4]. This latter factor, combined with a higher proportion of women in their sample, may explain the differences in comparison with our results [41].

Therefore, it is important to define a quantitative value that categorizes subjects with satisfactory or limited flexibility. Yıldırım et al. established cut-off values for diagnosing hamstring shortening: >33.0° for men and >23.4° for women [1]. In our sample, 39% of participants showed limitations in hamstring flexibility, which is lower than the 68% reported [4]. Due to the country-specific differences and the multifactorial nature of hamstring flexibility, direct comparisons remain challenging. However, both studies confirm that university students suffer from hamstring shortening, which could lead to musculoskeletal dysfunctions. Specifically, hamstring shortening is associated with a reduced ROM of the hip and/or knee, which can lead to biomechanical alterations, increase the risk of injuries, and negatively impact daily activities. Beyond flexibility, hamstring shortening not only affects joint mobility but also alters posture, which is associated with sciatic nerve stiffness [2]. As expected, a strong positive correlation was found between AKE and ASLR (r = 0.813, *p* < 0.001), reflecting their shared dependence on posterior chain flexibility. While this relationship may appear self-evident, it serves as internal validation of the consistency between the two commonly used functional tests of posterior chain mobility.

### 4.4. Limitations and Future Research

Several limitations should be acknowledged in our study. First, the ASLR test requires activation of the hip flexors, such as the iliopsoas, with an intensity of 40–60% of its maximum voluntary contraction (MVC). Previous research has reported activation greater than 60% MVC during the ASLR at approximately 60° of hip flexion, and between 40 and 60% MVC in the mid-range of the test, around 45° of hip flexion. This muscle activation may influence the obtained values, potentially interfering with the isolated measurement of sciatic nerve stiffness [38].

Although all participants engaged in at least two training sessions per week, specific data regarding the type, intensity, and duration of physical activity, as well as sedentary behavior, were not collected. Future studies should consider quantifying these variables given their potential impact on nerve stiffness and posterior chain flexibility.

Regarding ultrasound evaluation using SWE, sciatic nerve stiffness was assessed at a single location. Although this site was selected for its optimal imaging conditions, regional variations in nerve stiffness may exist along the sciatic nerve trajectory. Future studies should include multi-site assessments [8]. Several technical factors could have influenced the SWE measurements obtained in our study. Although minimal probe pressure and a standardized 2 mm ROI were used, small variations in probe compression could affect stiffness values. Additionally, inter-device variability is a known limitation of SWE, as different ultrasound systems and software algorithms may produce divergent results. Furthermore, all measurements were performed in a controlled laboratory environment at room temperature (22–24 °C), but future studies should monitor and report temperature conditions more precisely, given their potential impact on tissue stiffness.

Future studies should consider evaluating other structures in different anatomical locations or functional variables to establish a more precise correlation between SWE and ROM. Given that flexibility is influenced by multiple factors, it would be beneficial to explore how these various biomechanical components contribute to sciatic nerve stiffness and overall mobility. Moreover, our sample showed a male predominance (67.3%), which may have influenced our results, since previous studies have reported greater nerve stiffness and reduced posterior chain flexibility in males compared to females [42]. Therefore, future studies should aim for a balanced sex distribution to better understand potential sex-related differences in sciatic nerve stiffness and flexibility.

Finally, future studies should implement experimental designs to evaluate medium- and long-term alterations in sciatic nerve stiffness following modulation of either the sciatic nerve or musculoskeletal components of the posterior chain in populations with mobility restrictions. Additionally, the observed tendency between SWS and the AKE test indicates that studies with larger sample sizes may help clarify the potential relationship between SWE parameters and functional measures of posterior chain stiffness.

## 5. Practical Applications

The findings of this study provide reference values for sciatic nerve stiffness and posterior chain flexibility in healthy individuals. This normative framework may serve as a clinical reference for setting treatment goals and monitoring rehabilitation progress.

Although no correlation was observed between neural stiffness and flexibility, the increased SWE stiffness in individuals with reduced hamstring flexibility (*p* = 0.05) suggests that interventions aimed at improving neural mobility could positively influence range of motion.

From a clinical perspective, these results underscore the importance of incorporating neural tissue assessments using shear wave elastography alongside functional evaluations such as ROM, strength testing, or neurodynamic assessments.

## 6. Conclusions

Although no statistically significant correlations were observed between functional flexibility measures and SWE parameters, a group-based difference was identified, suggesting that SWE may detect structural changes not linearly reflected in ROM values. Individuals with greater limitation in posterior chain flexibility have demonstrated increased sciatic nerve stiffness, as measured by SWE. These findings highlight the multifactorial nature of flexibility and support the use of SWE as a complementary tool in the assessment of posterior chain restrictions.

## Figures and Tables

**Figure 1 sensors-25-02885-f001:**
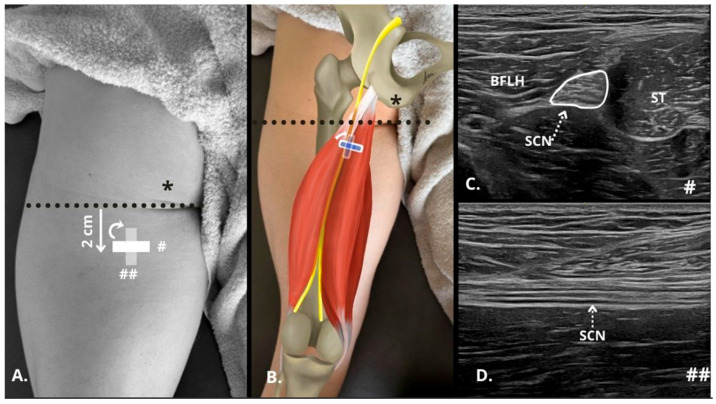
Ultrasonographic identification in B-mode of the sciatic nerve in the posterior thigh. (**A**,**B**) Diagram of transducer placement 2 cm from the gluteal fold (dashed black line) showing the positioning of the probe in the short axis (#) and long axis (##), taking as reference the position of the ischial tuberosity as the insertion of the hamstrings in the ultrasound examination (*). (**C**) Short axis of the sciatic nerve (SCN, perimeter marked with white line and dashed white line indicating the nerve) between the long head of the biceps femoris (BFLH) and the semitendinosus (ST). (**D**) Long axis of the SNC below (dashed white line indicating the nerve) the BFLH and ST. BFLH = long head of the biceps femoris; SCN = sciatic nerve; ST = semitendinosus.

**Figure 2 sensors-25-02885-f002:**
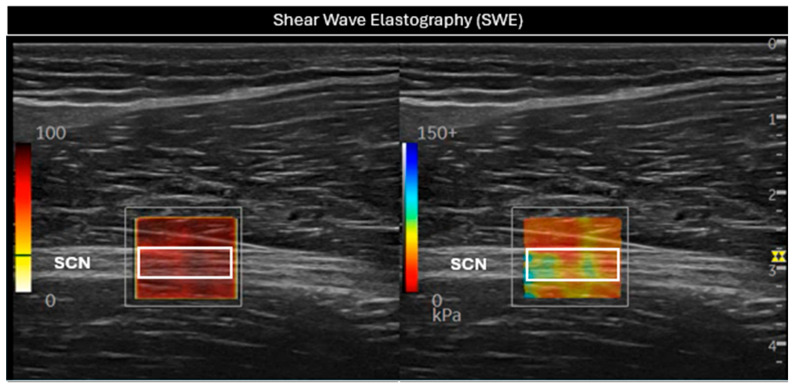
Evaluation of the sciatic nerve (SCN) in the long axis using shear wave elastography (SWE). In the (**left**) image, a white rectangular Q-box can be seen within the SWE P9R3 elastography system window, used to measure the SWS. In the (**right**) image, the stiffness of the nerve was assessed through the white rectangular Q-box. SCN = sciatic nerve.

**Figure 3 sensors-25-02885-f003:**
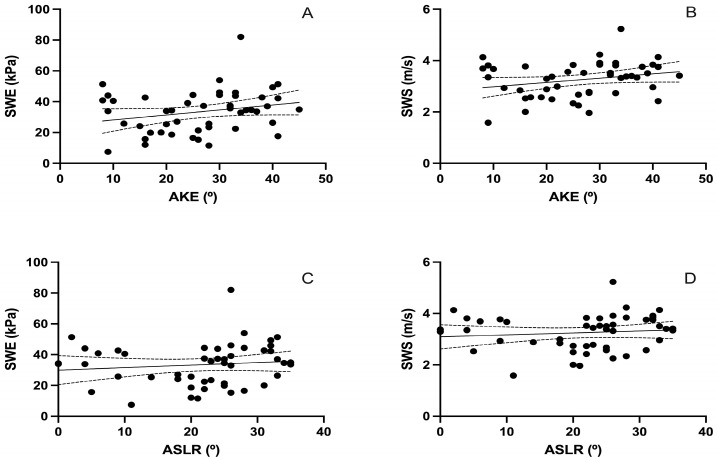
Correlation between posterior chain flexibility (AKE and ASLR) and sciatic nerve stiffness measured by SWE. (**A**,**B**) show scatter plots between the Active Knee Extension (AKE) test and sciatic nerve stiffness, measured as Young’s modulus in kilopascals (kPa) and shear wave speed (SWS) in meters per second (m/s), respectively. (**C**,**D**) show the corresponding correlations with the Active Straight Leg Raise (ASLR) test. Solid lines represent the linear regression fits; dashed lines indicate 95% confidence intervals. Each black dot represents an individual participant’s paired values for the two variables analyzed in each plot.

**Table 1 sensors-25-02885-t001:** Descriptive characteristics of the sample.

Variable	Total Sample (*N* = 25)
Age (years)	21.57 ± 2.34
Height (m)	1.77 ± 0.11
Weight (kg)	74.08 ± 15.48
BMI (kg/m^2^)	23.39 ± 2.61
Sex (male), n (%)	33 (67.3)
Limb Dominance (right), n (%)	21 (88)

Abbreviation: BMI, body mass index.

**Table 2 sensors-25-02885-t002:** Differences between the dominant and non-dominant leg.

Variable	Dominant Leg (n = 25)	Non-Dominant Leg (n = 24)	Mean Difference (95% CI)	*p*-Value	Cohen’s *d*
AKE (°)	27.20 ± 10.50	25.46 ± 10.35	1.74 (−4.25, 7.73)	0.28	0.17
ASLR (°)	21.48 ± 10.14	21.29 ± 10.05	0.18 (−5.61, 5.99)	0.47	0.02
SWE (kPa)	32.83 ± 15.53	34.09 ± 11.81	−1.26 (−9.21, 6.69)	0.38	−0.09
SWS (m/s)	3.21 ± 0.76	3.29 ± 0.62	−0.08 (−0.48, 0.32)	0.34	−0.12

Abbreviations: AKE, Active Knee Extension test; ASLR, Active Straight Leg Raise test; SWE, shear wave elastography; SWS, shear wave speed.

**Table 3 sensors-25-02885-t003:** Differences between limited and adequate hamstring stiffness groups.

Variable	Limited (n = 19)	Adequate (n = 30)	Mean Difference (95% CI)	*p*-Value	Cohen’s *d*
AKE (°)	35.68 ± 5.50	20.43 ± 8.08	15.25 (11.00, 19.50)	**<0.001**	2.12
ASLR (°)	29.16 ± 4.49	16.47 ± 9.39	12.69 (8.04, 17.34)	**<0.001**	1.61
SWE (kPa)	38.18 ± 14.45	30.45 ± 12.55	7.72 (−0.12, 15.58)	**0.05**	0.61
SWS (m/s)	3.49 ± 0.64	3.10 ± 0.68	0.39 (−0.01, 0.78)	**0.05**	0.60

Abbreviations: AKE, Active Knee Extension test; ASLR, Active Straight Leg Raise test; SWE, shear wave elastography; SWS, shear wave speed; Statistical significance was set at *p* < 0.05, corresponding to a 95% confidence interval. Significant results are highlighted in bold.

**Table 4 sensors-25-02885-t004:** Correlation matrix between flexibility and SWE parameters.

Variable	AKE (°)	ASLR (°)	SWE (kPa)
1. AKE (°)			
2. ASLR (°)	0.813 (<0.001)		
3. SWE (kPa)	0.245 (0.089)	0.119 (0.416)	
4. SWS (m/s)	0.251 (0.082)	0.110 (0.453)	0.986 (<0.001)

Abbreviations: AKE, Active Knee Extension test; ASLR, Active Straight Leg Raise test; SWE, shear wave elastography; SWS, shear wave speed. Data are presented as r (*p*-value).

## Data Availability

The data presented in this study are available on request from the corresponding author.

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
