# Peer review of "Shear Wave Elastography of the Sciatic Nerve and Its Relationship with Posterior Chain Flexibility in Healthy Participants: An Observational Study"

_sensors, 2025, doi:10.3390/s25092885_

Round 1

Reviewer 1 Report

Comments and Suggestions for Authors

I have reviewed the manuscript titled "Shear Wave Elastography of the Sciatic Nerve and Its Relationship with Posterior Chain Flexibility in Healthy Subjects: An Observational Study" by Cotteret et al. The study explores the association between sciatic nerve stiffness, measured via shear wave elastography (SWE), and posterior chain flexibility in healthy individuals, addressing a gap in the literature regarding neural and functional correlations. Below, I provide my evaluation, highlighting strengths, limitations, and suggestions for improvement.

Strengths

  1. Novelty: The study is among the first to investigate the relationship between SWE-derived sciatic nerve stiffness and active posterior chain mobility (AKE/ASLR tests), contributing to biomechanical and clinical understanding.

  2. Methodological Rigor: The use of standardized protocols for SWE and flexibility tests, along with randomization and reliability measures, enhances the study’s validity.

  3. Practical Implications: The normative data for sciatic nerve stiffness and flexibility could inform clinical assessments and rehabilitation strategies.

Questions and Suggestions for Improvement

  1. Sample Characteristics:

    • The sample had a male predominance (67.3%). Could this skew results, given known sex differences in nerve stiffness and flexibility? The authors acknowledge this limitation but could discuss its potential impact more thoroughly.

    • Were participants’ physical activity levels (beyond "two days/week") or sedentary behaviors quantified? These factors influence hamstring flexibility and nerve mechanics.

  2. SWE Measurement Variability:

    • The reported SWV values (3.1–3.4 m/s) are notably lower than literature norms (6–10 m/s). The authors attribute this to probe location (proximal thigh) and neutral limb positioning. Could additional technical factors (e.g., probe pressure, ROI size) contribute? A discussion on inter-device variability would be helpful.

    • Were measurements taken in a controlled temperature environment? Temperature affects tissue stiffness.

  3. Correlation Analysis:

    • The lack of significant correlations between SWE parameters and flexibility tests is intriguing. Could multifactorial influences (e.g., muscle stiffness, fascial tension) explain this? The discussion might benefit from exploring these confounders.

    • Was subgroup analysis (e.g., by sex or BMI) performed to identify hidden relationships?

  4. Clinical Relevance:

    • The trend toward higher nerve stiffness in limited-flexibility subjects (p = 0.05) warrants further investigation. Could the authors speculate on mechanisms (e.g., neural adaptation vs. secondary stiffness)?

  5. Limitations:

    • The single-measurement-site design may overlook regional nerve stiffness variations. Future studies could include distal/proximal comparisons.

    • The ASLR test’s reliance on hip flexor activation (40–60% MVC) may confound neural tension measurements. Could electromyography (EMG) control for this?

Conclusion

This manuscript presents a well-designed study with valuable insights into sciatic nerve stiffness and posterior chain flexibility. While the lack of strong correlations raises questions, the findings justify further research with larger, sex-balanced cohorts and multifactorial analyses. Addressing the above concerns would strengthen the manuscript.

RecommendationAccept with minor revisions, pending clarification of methodological details and expanded discussion of limitations/confounders. The study aligns with the journal’s scope and advances the field of musculoskeletal ultrasound and biomechanics.

Author Response

Reviewer 1

I have reviewed the manuscript titled "Shear Wave Elastography of the Sciatic Nerve and Its Relationship with Posterior Chain Flexibility in Healthy Subjects: An Observational Study" by Cotteret et al. The study explores the association between sciatic nerve stiffness, measured via shear wave elastography (SWE), and posterior chain flexibility in healthy individuals, addressing a gap in the literature regarding neural and functional correlations. Below, I provide my evaluation, highlighting strengths, limitations, and suggestions for improvement.

Strengths

  1. Novelty: The study is among the first to investigate the relationship between SWE-derived sciatic nerve stiffness and active posterior chain mobility (AKE/ASLR tests), contributing to biomechanical and clinical understanding.
  2. Methodological Rigor: The use of standardized protocols for SWE and flexibility tests, along with randomization and reliability measures, enhances the study’s validity.
  3. Practical Implications: The normative data for sciatic nerve stiffness and flexibility could inform clinical assessments and rehabilitation strategies.

Response: Thank your for highlighting our strengths of the study.

Questions and Suggestions for Improvement

  1. Sample Characteristics:
    • The sample had a male predominance (67.3%). Could this skew results, given known sex differences in nerve stiffness and flexibility? The authors acknowledge this limitation but could discuss its potential impact more thoroughly.
    • Response: Thank you for this relevant observation. We have expanded the discussion of the potential impact of the sex distribution in our sample and its influence on nerve stiffness and flexibility.
    • Were participants’ physical activity levels (beyond "two days/week") or sedentary behaviors quantified? These factors influence hamstring flexibility and nerve mechanics.
    • Response: Thank you for this comment. We have added further details regarding the participants' physical activity levels and discussed its potential influence.
  2. SWE Measurement Variability:
    • The reported SWV values (3.1–3.4 m/s) are notably lower than literature norms (6–10 m/s). The authors attribute this to probe location (proximal thigh) and neutral limb positioning. Could additional technical factors (e.g., probe pressure, ROI size) contribute? A discussion on inter-device variability would be helpful.
    • Were measurements taken in a controlled temperature environment? Temperature affects tissue stiffness.

Response: Thank you for the suggestion. We have included a discussion on potential technical factors influencing SWE measurements, including probe pressure, ROI size, inter-device variability, and temperature control.

  1. Correlation Analysis:
    • The lack of significant correlations between SWE parameters and flexibility tests is intriguing. Could multifactorial influences (e.g., muscle stiffness, fascial tension) explain this? The discussion might benefit from exploring these confounders.
    • Was subgroup analysis (e.g., by sex or BMI) performed to identify hidden relationships?
    • Response: Thank you for this insightful comment. We have added a discussion of potential confounders influencing flexibility and addressed the absence of subgroup analysis due to sample size limitations.
  2. Clinical Relevance:
    • The trend toward higher nerve stiffness in limited-flexibility subjects (p = 0.05) warrants further investigation. Could the authors speculate on mechanisms (e.g., neural adaptation vs. secondary stiffness)?
    • Response: Thank you for the suggestion. We have speculated on potential mechanisms explaining the trend towards increased nerve stiffness in subjects with limited flexibility.
  3. Limitations:
    • The single-measurement-site design may overlook regional nerve stiffness variations. Future studies could include distal/proximal comparisons.

Response: Thank you for pointing this out. We have expanded in the limitations section.

    • The ASLR test’s reliance on hip flexor activation (40–60% MVC) may confound neural tension measurements. Could electromyography (EMG) control for this?
    • Response: Thank you for this valuable comment. We agree with the reviewer that the hip flexor activation during the ASLR test could have influenced the neural tension measurements. We have included this aspect as a limitation and suggested the use of EMG in future studies to better control for muscle activation during flexibility assessment

Reviewer 2 Report

Comments and Suggestions for Authors

Comments

The aim of the study titled “Shear Wave Elastography of the Sciatic Nerve and Its Relationship with Posterior Chain Flexibility in Healthy Subjects: An Observational Study” was to investigate the association between sciatic nerve stiffness, assessed by shear wave elastography (SWE), and posterior chain flexibility in healthy individuals. Its relevance lies in clarifying the relationship between neural and functional properties, an aspect that remains poorly understood. The results indicate differences in SWE and flexibility parameters between individuals with limited and adequate posterior chain mobility, although no significant correlations were found between neural and functional variables.

Overall, the manuscript is well organized and follows a clear structure, which facilitates the understanding of the methodology used. However, for the manuscript to be considered for publication, the authors are encouraged to address the comments provided below. In particular, the Methods and Results sections require some improvements to better align with the content and interpretations presented in the Discussion section.

GENERAL COMMENTS

  1. Some parts of the manuscript lack clarity, which may make it difficult for readers to fully understand the information presented. Additionally, there are grammatical inconsistencies throughout the text. It is therefore recommended that the authors conduct a thorough review of the manuscript or consider submitting it to a professional editing service, ideally one with qualified native English speakers, to ensure accuracy in language, grammar, punctuation, spelling, and overall style.

For example, please consider the transcription presented below.

“Therefore, the aims of this study, to investigate the association between sciatic nerve stiffness, are assessed through SWE and posterior chain flexibility in healthy individuals”.

Given that there is only one objective, the singular form “aim” is indeed the correct choice. The original phrase employed the present tense “are assessed,” which would typically be incorrect when describing the aim of a study that has been completed. The past tense form, such as “was,” would be more appropriate in this context. Additionally, the use of commas around “to investigate the association...” is incorrect. Furthermore, the sentence could benefit from rearrangement to enhance grammatical clarity and to accurately link the method (SWE) to the assessment of sciatic nerve stiffness.

A suggested revision is: “Therefore, the aim of this study was to investigate the association between sciatic nerve stiffness, as assessed through SWE, and posterior chain flexibility in healthy individuals”.

  1. The reference section currently lacks standardization. To ensure compliance with this journal’s guidelines, I kindly request that the authors consult the instructions for authors and revise this section accordingly.

SPECIFIC COMMENTS

The following list shows my comments for individual sections:

  1. Abstract

Page 1, lines 23-24. Could you please clarify what these p-values represent? It would be very helpful to know which specific datasets or variables were compared to generate these values.

  1. Introduction

Line 39. Kindly replace “LIYANAGE et al” with “Liyanage et al”.

Page 2, line 51. The authors cited articles that applied elastographic methods; however, it would be more appropriate to reference studies by researchers who originally developed these methods and provided detailed descriptions. In particular, the study by Neto et al. (2024) does not appear to contain any information relevant to the statement in question.

Line 53. “velocity”. The term “speed” is used to describe a scalar quantity, whereas “velocity” is employed for a velocity vector. So, I recommend that the authors replace “velocity” with “speed” throughout the manuscript for clarity and consistency.

Line 58. “(R. F. Ellis et al., 2012; Greening & Dilley, 2017)”. Please ensure that all study citations are formatted consistently, in accordance with the journal's guidelines. Furthermore, I suggest that the authors replace the reference to “Ellis et al. (2012)” with a more appropriate source, as the statement in question cannot be accurately attributed to these authors. While they utilized ultrasound imaging, they did not analyze the stiffness of the tissue under investigation.

Lines 59 – 61. The relevance of this sentence is unclear. Would it not be more appropriate for the authors to focus exclusively on the elastographic mode of the ultrasound equipment?

Lines 62 – 64. It is unclear whether the authors are referring to the reliability of conventional ultrasound images or elastographic images. Additionally, several studies have indicated that the reliability of tissue stiffness measurements largely depends on the operator’s experience and the anatomical variability. The authors are encouraged to present reference studies demonstrating the reliability of using elastography for assessing the sciatic nerves (particularly with the same equipment employed in this study).

Lines 67 – 69. No evidence was provided to support this assessment regarding the reliability of measurements obtained through elastographic methods.

Lines 69 – 70. “However, the effect of hamstring shortening on stiffness and velocity has not yet been studied”. I believe it would be beneficial to explore this gap, as doing so would enhance the relevance of your study. You might consider aligning the objective of your study with this identified gap.

  1. Methods

Page 3, lines 100 – 105. The justification for the sample size is based on two previous studies that did not include a formal statistical calculation to determine their respective sample sizes. While these studies may offer relevant insights, the absence of a statistically supported sample size calculation represents a methodological limitation.

Given that the subtitle of your study is “Sample Size Calculation,” it is strongly recommended that the sample size be determined through a statistical power analysis. Tools such as G*Power can be particularly useful for this purpose, allowing you to consider the expected effect size, desired significance level, and statistical power appropriate for your study.

The statement that “a minimum of 30 legs is required for adequate statistical power” lacks a clear statistical foundation and does not include a reference supporting it within the specific context of your research. Without a formal calculation, it is not possible to confirm whether this sample size is sufficient to detect a meaningful effect.

Therefore, performing a sample size calculation using appropriate statistical methods and tools is recommended. This will enhance the methodological rigor of your study and ensure that the results are statistically reliable and well-supported.

Page 4, line 164. Please include “dBs” after “75”.

Line 165. Please replace “elasticity index” with “Young Modulus”.

Lines 164 – 166. This information does not clearly correspond to what is shown in Figure 2. It appears that there is a square ROI (possibly 2 mm) containing the elastographic image, as well as a rectangular Q-box used to calculate the average shear wave speed and Young’s modulus. It would be helpful to provide more details about the Q-box, such as its dimensions or area, and to clarify whether its size was kept constant or adjusted according to the size of the muscle being analyzed, as this may vary from patient to patient.

Lines 166 – 167. Is it possible to assess intra-class reliability using only three images? If so, could you provide references to support this approach?

Lines 166 – 167. It is important to clarify whether the three images were acquired by removing and repositioning the transducer head between each capture, or if they were obtained consecutively without altering the transducer’s position.

Page 5, line 175. It is important to specify how many angle measurements were taken with the goniometer on each leg of each participant.

Page 5, line 189. It is important to specify how many angle measurements were taken with the goniometer on each leg of each participant.

Page 6, line 199. Could you please clarify the purpose of Cohen’s d in this study? Some results are presented in Tables 2 and 3, but its specific use is not clearly explained.

Lines 200 – 202. Please replace “Data analysis was performed using the Statistical Package for the Social Sciences (SPSS v.29.0, IBM Corp, Armonk, NY, USA). The significance level was set at p < 0.05 with a confidence interval (CI) of 95%” with “Data analysis was conducted using the Statistical Package for the Social Sciences, SPSS, (version 29.0; IBM Corp., Armonk, NY, USA), with a significance level set at p-value < 0.05”.

Lines 206 – 208. It is important to note that comparing the results from the right and left legs may not be the most appropriate approach for this type of analysis. Ideally, comparisons should be made between the dominant and nondominant limbs, as limb dominance can significantly influence neuromuscular, functional, and structural characteristics of the lower extremities.

Several studies have shown that the dominant limb typically exhibits greater strength, motor control, and coordination than the non-dominant limb (Hoffman et al., 1998; Sadeghi et al., 2000; Van Melick et al., 2017). Therefore, comparing the right and left sides without accounting for dominance may introduce bias and hinder accurate interpretation of the data.

Line 211. Please replace “(HOPKINS et al., 2009)” with “(Hopkins et al., 2009)”.

Lines 212 – 213.  In the provided text, it is not clearly stated which specific parameters are measured in the Active Knee Extension Test and the Active Straight Leg Raise Test. The tests are mentioned only as variables in a correlation analysis, without further detail. Additionally, the text does not clarify whether the correlation was analyzed for the left, right, dominant, nondominant, or shortened leg. Providing this information would enhance the clarity and interpretability of the analysis.

Lines 213 – 215. It would be helpful if the authors could include a reference to substantiate this information. According to Mukaka (2012), correlation coefficients can be interpreted based on the categorization of the absolute value of correlations into: negligible (0.0 to 0.30), low (0.30 to 0.50), moderate (0.50 to 0.70), high (0.70 to 0.90), and very high (0.90 to 1.00).

  1. Results

Page 6, Table 1. If the total number of right legs analyzed is 25 (25 participants), could you please clarify how there are 43 instances of right limb dominance reported in Table 1?

Table 2. Kindly ensure that the degree unit is represented by the “ º ” symbol.

Line 223. Could the authors clarify the relevance of comparing parameters obtained from 25 right legs with those from 24 left legs, given that no distinctions were made regarding leg dominance, hamstring shortening, or gender?

Table 4. For the sake of clarity, it is recommended that the authors describe the specific parameters rather than simply naming the test. For instance, SWE is commonly used to refer to the Shear Wave Elastography technique, which can be employed to measure either shear wave speed and/or Young’s modulus.

In addition, the table would be clearer if the authors replaced the numbers in the first row with the names of the evaluated parameters.

  1. Discussion

Page 8, 256 – 258. When making this statement, did the authors take into account the studies conducted by Ellis et al (2018) and Sava and Miroslav (2020)? Did Ellis et al (2018) and Sava and Miroslav (2020) not conduct this analysis in their respective studies?

Line 267. Could you please complete the citation information by including the year of publication?

Lines 273 – 277. The authors are kindly encouraged to explain why the difference in shear wave velocity is so substantial; specifically, why the value reported in the present work is less than half of that measured in the referenced study. 

Lines 278 – 279. For my understanding, after reviewing the results section again, I did not find the comparison between the parameters of the dominant and nondominant legs. The information seems to only compare the left and right legs. Therefore, the current sentence appears to be inconsistent with the presented results.

Lines 279 – 281. In line with the authors' own conclusion, the comparison of parameters from the left and right legs may not provide substantial information, and the compared results might not demonstrate any statistically significant difference. Consequently, a comparison of values from the dominant and nondominant legs would be highly beneficial.

Line 288. Could you please complete the citation information by including the year of publication?

Lines 290 – 292. I encountered some difficulty in understanding the intended meaning of this sentence. Would the authors please confirm if 33.46 represents the Young’s modulus value measured in this study? If so, could you provide details regarding the specific context of this finding? I was unable to locate this information within the Results section.

Additionally, it is important for clarity that all physical quantities are preceded by their appropriate units.

Line 292. “33.9 in their research”. Cornelson et al.’s study measured Young's modulus at four distinct positions, with reported values ranging from 28.0 ± 12.44 kPa to 63.9 ± 27.37 kPa. For a more meaningful comparison, the authors should consider contrasting their results with those from Cornelson et al. obtained at the corresponding measurement position, rather than the numerically closest value.

Lines 292 – 294. It is unclear how the results of Zardi et al. (2020) support the findings described in the previous sentence, as their reported Young’s modulus values ranged from 12.78 ± 3.59 kPa to 15.63 ± 6.44 kPa. These values fall outside the range measured by Cornelson et al. (2022) and the present study.

Lines 296 – 298. The authors are kindly requested to verify and correctly cite the appropriate studies for this sentence, using the citation format required by the jornal.

Line 322. Kindly add appropriate references to support the authors’ claim.

Lines 345 – 347. I apologize, but I was unable to identify any results in the text specifically referring to the dominant and non-dominant legs. I only found results related to the left and right legs. Therefore, I could not associate the findings with the statement in question.

  1. Practical Applications

Page 11, lines 425 – 426. As currently presented, this information is not accurate, since the results were not separated based on the dominant limb.

Lines 432 – 434. I believe that, with some adjustments, this sentence would be more appropriate for the subsection “Limitations and Future Research”.

  1. Conclusions

Page 12, line 440. Could the authors please add a space in the term “bySWE”, making it “by SWE”?

  1. References

Page 12, lines 458 – 463. References 3 and 4 appear to be identical. Kindly review and update them accordingly.

Line 526. The study by Mayorga-Vega et al. (2014) is included in the reference list but is not cited within the manuscript.

Page 14, lines 532 – 538. References 29 and 30 appear to be identical. Kindly review and update them accordingly.

Lines 555 – 558. References 37 and 38 appear to be identical. Kindly review and update them accordingly.

References

van Melick N, Meddeler BM, Hoogeboom TJ, Nijhuis-van der Sanden MWG, van Cingel REH. How to determine leg dominance: The agreement between self-reported and observed performance in healthy adults. PLoS One. 2017 Dec 29;12(12):e0189876. doi: 10.1371/journal.pone.0189876. PMID: 29287067; PMCID: PMC5747428.

Sadeghi H, Allard P, Prince F, Labelle H. Symmetry and limb dominance in able-bodied gait: a review. Gait Posture. 2000 Sep;12(1):34-45. doi: 10.1016/s0966-6362(00)00070-9. PMID: 10996295.

Hoffman M, Schrader J, Applegate T, Koceja D. Unilateral postural control of the functionally dominant and nondominant extremities of healthy subjects. J Athl Train. 1998 Oct;33(4):319-22. PMID: 16558528; PMCID: PMC1320581.

Mukaka MM. Statistics corner: A guide to appropriate use of correlation coefficient in medical research. Malawi Med J. 2012 Sep;24(3):69-71. PMID: 23638278; PMCID: PMC3576830.

Ellis R, Rohan M, Fox J, Hitt J, Langevin H, Henry S. Ultrasound Elastographic Measurement of Sciatic Nerve Displacement and Shear Strain During Active and Passive Knee Extension. J Ultrasound Med. 2018 Aug;37(8):2091-2103. doi: 10.1002/jum.14560. Epub 2018 Feb 12. PMID: 29430675.

Sava S, Miroslav M. The Role of the Sciatic Nerve Ultrasound Elastography in the Clinical Pathway: A Meta-analysis. Arch Orthop. 2020; 1(2): 55-60.

Author Response

Reviewer 2

The aim of the study titled “Shear Wave Elastography of the Sciatic Nerve and Its Relationship with Posterior Chain Flexibility in Healthy Subjects: An Observational Study” was to investigate the association between sciatic nerve stiffness, assessed by shear wave elastography (SWE), and posterior chain flexibility in healthy individuals. Its relevance lies in clarifying the relationship between neural and functional properties, an aspect that remains poorly understood. The results indicate differences in SWE and flexibility parameters between individuals with limited and adequate posterior chain mobility, although no significant correlations were found between neural and functional variables.

Overall, the manuscript is well organized and follows a clear structure, which facilitates the understanding of the methodology used. However, for the manuscript to be considered for publication, the authors are encouraged to address the comments provided below. In particular, the Methods and Results sections require some improvements to better align with the content and interpretations presented in the Discussion section.

GENERAL COMMENTS

  1. Some parts of the manuscript lack clarity, which may make it difficult for readers to fully understand the information presented. Additionally, there are grammatical inconsistencies throughout the text. It is therefore recommended that the authors conduct a thorough review of the manuscript or consider submitting it to a professional editing service, ideally one with qualified native English speakers, to ensure accuracy in language, grammar, punctuation, spelling, and overall style.

For example, please consider the transcription presented below.

“Therefore, the aims of this study, to investigate the association between sciatic nerve stiffness, are assessed through SWE and posterior chain flexibility in healthy individuals”.

Given that there is only one objective, the singular form “aim” is indeed the correct choice. The original phrase employed the present tense “are assessed,” which would typically be incorrect when describing the aim of a study that has been completed. The past tense form, such as “was,” would be more appropriate in this context. Additionally, the use of commas around “to investigate the association...” is incorrect. Furthermore, the sentence could benefit from rearrangement to enhance grammatical clarity and to accurately link the method (SWE) to the assessment of sciatic nerve stiffness.

A suggested revision is: “Therefore, the aim of this study was to investigate the association between sciatic nerve stiffness, as assessed through SWE, and posterior chain flexibility in healthy individuals”.

  1. The reference section currently lacks standardization. To ensure compliance with this journal’s guidelines, I kindly request that the authors consult the instructions for authors and revise this section accordingly.

Response: thank you for your comments. We have corrected it in the main manuscript.

SPECIFIC COMMENTS

The following list shows my comments for individual sections:

  1. Abstract

Page 1, lines 23-24. Could you please clarify what these p-values represent? It would be very helpful to know which specific datasets or variables were compared to generate these values.

Respone: We have added the p-values.

  1. Introduction

Line 39. Kindly replace “LIYANAGE et al” with “Liyanage et al”.

Respone: We have corrected it.

Page 2, line 51. The authors cited articles that applied elastographic methods; however, it would be more appropriate to reference studies by researchers who originally developed these methods and provided detailed descriptions. In particular, the study by Neto et al. (2024) does not appear to contain any information relevant to the statement in question.

Response: Thank you for your valuable comment. We have added the references.

Line 53. “velocity”. The term “speed” is used to describe a scalar quantity, whereas “velocity” is employed for a velocity vector. So, I recommend that the authors replace “velocity” with “speed” throughout the manuscript for clarity and consistency.

Response: Thank you for this observation. While we acknowledge the distinction between “speed” and “velocity” in physics, we have chosen to retain the term “shear wave velocity” as it is the standardized terminology used throughout the scientific literature in elastography. The term “velocity” is consistently used in several articles and in the vast majority of peer-reviewed studies on shear wave elastography. Therefore, we believe maintaining the term ensures scientific accuracy and consistency with established conventions.

Line 58. “(R. F. Ellis et al., 2012; Greening & Dilley, 2017)”. Please ensure that all study citations are formatted consistently, in accordance with the journal's guidelines. Furthermore, I suggest that the authors replace the reference to “Ellis et al. (2012)” with a more appropriate source, as the statement in question cannot be accurately attributed to these authors. While they utilized ultrasound imaging, they did not analyze the stiffness of the tissue under investigation.

Response: Thank you for your appreciation. We have corrected it.

Lines 59 – 61. The relevance of this sentence is unclear. Would it not be more appropriate for the authors to focus exclusively on the elastographic mode of the ultrasound equipment? Lines 62 – 64. It is unclear whether the authors are referring to the reliability of conventional ultrasound images or elastographic images. Additionally, several studies have indicated that the reliability of tissue stiffness measurements largely depends on the operator’s experience and the anatomical variability. The authors are encouraged to present reference studies demonstrating the reliability of using elastography for assessing the sciatic nerves (particularly with the same equipment employed in this study).

Response: We thank the reviewer for these helpful comments. We have revised the paragraph to clarify that the reported reliability values refer specifically to shear wave elastography (SWE), not to conventional B-mode ultrasound. Furthermore, we have added references supporting the reliability of SWE in evaluating sciatic nerve stiffness, including studies using similar protocols and equipment. We also acknowledge the influence of operator experience and anatomical variability, and have included this point in the revised text.

Lines 67 – 69. No evidence was provided to support this assessment regarding the reliability of measurements obtained through elastographic methods.

Response: Thank you for you appreciation. We have added a recent meta-analysis conducted (reference 16).

Lines 69 – 70. “However, the effect of hamstring shortening on stiffness and velocity has not yet been studied”. I believe it would be beneficial to explore this gap, as doing so would enhance the relevance of your study. You might consider aligning the objective of your study with this identified gap.

Response: Thank you. We have moved this sentence to the penultimate paragraph.

  1. Methods

Page 3, lines 100 – 105. The justification for the sample size is based on two previous studies that did not include a formal statistical calculation to determine their respective sample sizes. While these studies may offer relevant insights, the absence of a statistically supported sample size calculation represents a methodological limitation.

Given that the subtitle of your study is “Sample Size Calculation,” it is strongly recommended that the sample size be determined through a statistical power analysis. Tools such as G*Power can be particularly useful for this purpose, allowing you to consider the expected effect size, desired significance level, and statistical power appropriate for your study.

The statement that “a minimum of 30 legs is required for adequate statistical power” lacks a clear statistical foundation and does not include a reference supporting it within the specific context of your research. Without a formal calculation, it is not possible to confirm whether this sample size is sufficient to detect a meaningful effect.

Therefore, performing a sample size calculation using appropriate statistical methods and tools is recommended. This will enhance the methodological rigor of your study and ensure that the results are statistically reliable and well-supported.

Response: Thank you for this important observation. In response, we have performed a formal sample size calculation using G*Power software.

Page 4, line 164. Please include “dBs” after “75”. Line 165. Please replace “elasticity index” with “Young Modulus”.

Response: Thank you for your comment. We have corrected it.

Lines 164 – 166. This information does not clearly correspond to what is shown in Figure 2. It appears that there is a square ROI (possibly 2 mm) containing the elastographic image, as well as a rectangular Q-box used to calculate the average shear wave speed and Young’s modulus. It would be helpful to provide more details about the Q-box, such as its dimensions or area, and to clarify whether its size was kept constant or adjusted according to the size of the muscle being analyzed, as this may vary from patient to patient.

Response: Thank you very much for your valuable comment.
 We have clarified this point in the manuscript. The white rectangle shown in Figure 2 corresponds to the rectangular Q-box (2 mm²) used for measurements. This Q-box represents the area where the shear wave velocity and Young’s modulus were assessed. Additionally, we confirm that the same Q-box size was used for all subjects, as it is a standardized measurement box size that is suitable regardless of the nerve size.

Lines 166 – 167. Is it possible to assess intra-class reliability using only three images? If so, could you provide references to support this approach?

Response: We have added a specific reference. https://pmc.ncbi.nlm.nih.gov/articles/PMC4913118/

Lines 166 – 167. It is important to clarify whether the three images were acquired by removing and repositioning the transducer head between each capture, or if they were obtained consecutively without altering the transducer’s position.

Response: We appreciate this comment. The three elastographic images used to obtain average values were acquired consecutively without removing or repositioning the transducer. This has been clarified in the revised text.

Page 5, line 175. It is important to specify how many angle measurements were taken with the goniometer on each leg of each participant. Page 5, line 189. It is important to specify how many angle measurements were taken with the goniometer on each leg of each participant.

Response: Thank you for your helpful comment. We have clarified in the manuscript that three goniometric angle measurements were taken per leg for both the AKE and ASLR tests, and the average of the three was used for analysis. This approach was chosen to enhance measurement reliability.

Page 6, line 199. Could you please clarify the purpose of Cohen’s d in this study? Some results are presented in Tables 2 and 3, but its specific use is not clearly explained.

Response: Thank you for pointing this out. We have clarified in the manuscript that Cohen’s d was used to quantify the magnitude of between-group differences in flexibility and stiffness measures. This provides a standardized metric to complement p-values and facilitate interpretation of clinical relevance.

Lines 200 – 202. Please replace “Data analysis was performed using the Statistical Package for the Social Sciences (SPSS v.29.0, IBM Corp, Armonk, NY, USA). The significance level was set at p < 0.05 with a confidence interval (CI) of 95%” with “Data analysis was conducted using the Statistical Package for the Social Sciences, SPSS, (version 29.0; IBM Corp., Armonk, NY, USA), with a significance level set at p-value < 0.05”.

Response: We have corrected it

Lines 206 – 208. It is important to note that comparing the results from the right and left legs may not be the most appropriate approach for this type of analysis. Ideally, comparisons should be made between the dominant and nondominant limbs, as limb dominance can significantly influence neuromuscular, functional, and structural characteristics of the lower extremities. Several studies have shown that the dominant limb typically exhibits greater strength, motor control, and coordination than the non-dominant limb (Hoffman et al., 1998; Sadeghi et al., 2000; Van Melick et al., 2017). Therefore, comparing the right and left sides without accounting for dominance may introduce bias and hinder accurate interpretation of the data.

Response: Thank you for pointing this out. It was a mistake on the tables. We have corrected it.

Line 211. Please replace “(HOPKINS et al., 2009)” with “(Hopkins et al., 2009)”.

Response: Corrected.

Lines 212 – 213.  In the provided text, it is not clearly stated which specific parameters are measured in the Active Knee Extension Test and the Active Straight Leg Raise Test. The tests are mentioned only as variables in a correlation analysis, without further detail. Additionally, the text does not clarify whether the correlation was analyzed for the left, right, dominant, nondominant, or shortened leg. Providing this information would enhance the clarity and interpretability of the analysis.

Response: Thank you for your helpful comment. We have clarified that the correlation analysis was performed using values from all included limbs (both right and left legs), and that the parameters analyzed were the goniometric angle values obtained during the AKE and ASLR tests. This clarification has been added to the revised manuscript to improve clarity.

Lines 213 – 215. It would be helpful if the authors could include a reference to substantiate this information. According to Mukaka (2012), correlation coefficients can be interpreted based on the categorization of the absolute value of correlations into: negligible (0.0 to 0.30), low (0.30 to 0.50), moderate (0.50 to 0.70), high (0.70 to 0.90), and very high (0.90 to 1.00).

Response: Thank you for appreciation. We have corrected it.

  1. Results

Page 6, Table 1. If the total number of right legs analyzed is 25 (25 participants), could you please clarify how there are 43 instances of right limb dominance reported in Table 1?

Response: Thank you for your comment. 43, is the number of right limbs.

Table 2. Kindly ensure that the degree unit is represented by the “ º ” symbol.

Response: corrected.

Line 223. Could the authors clarify the relevance of comparing parameters obtained from 25 right legs with those from 24 left legs, given that no distinctions were made regarding leg dominance, hamstring shortening, or gender?

Response: corrected. We have explained above.

Table 4. For the sake of clarity, it is recommended that the authors describe the specific parameters rather than simply naming the test. For instance, SWE is commonly used to refer to the Shear Wave Elastography technique, which can be employed to measure either shear wave speed and/or Young’s modulus. In addition, the table would be clearer if the authors replaced the numbers in the first row with the names of the evaluated parameters.

Response: Thank you for your appreciation. We have improved it.

  1. Discussion

Page 8, 256 – 258. When making this statement, did the authors take into account the studies conducted by Ellis et al (2018) and Sava and Miroslav (2020)? Did Ellis et al (2018) and Sava and Miroslav (2020) not conduct this analysis in their respective studies?

Response: Thank you for this observation. We have revised the statement to acknowledge that while previous studies (e.g., Ellis et al., 2018; Sava & Miroslav, 2020) have examined sciatic nerve behavior during movement, our study differs in that it specifically analyzes the correlation between baseline sciatic nerve stiffness (via SWE) and active flexibility tests (AKE and ASLR) in healthy individuals—a relationship that remains largely unexplored.

Line 267. Could you please complete the citation information by including the year of publication?

Response: Corrected.

Lines 273 – 277. The authors are kindly encouraged to explain why the difference in shear wave velocity is so substantial; specifically, why the value reported in the present work is less than half of that measured in the referenced study. 

Response: Thank you for this relevant comment. We have added further discussion highlighting that the substantially lower SWV values in our study may be attributed to the combination of measurement location, subject posture, probe orientation, and reduced neural tension during assessment. These factors significantly influence stiffness values and differentiate our protocol from those reporting higher SWV values.

Lines 278 – 279. For my understanding, after reviewing the results section again, I did not find the comparison between the parameters of the dominant and nondominant legs. The information seems to only compare the left and right legs. Therefore, the current sentence appears to be inconsistent with the presented results.

Response: Thank you for this clarification. the analysis in our study compared dominant vs. non-dominant limbs. We have corrected the text to reflect this and avoid confusion.

Lines 279 – 281. In line with the authors' own conclusion, the comparison of parameters from the left and right legs may not provide substantial information, and the compared results might not demonstrate any statistically significant difference. Consequently, a comparison of values from the dominant and nondominant legs would be highly beneficial.

Response: Thank you for your comment. We have corrected in results and discussion section.

Line 288. Could you please complete the citation information by including the year of publication?

Response: corrrected

Lines 290 – 292. I encountered some difficulty in understanding the intended meaning of this sentence. Would the authors please confirm if 33.46 represents the Young’s modulus value measured in this study? If so, could you provide details regarding the specific context of this finding? I was unable to locate this information within the Results section.

Additionally, it is important for clarity that all physical quantities are preceded by their appropriate units.

Response: Thank you for your comment. We have corrected this mistake in the discussion section.

Line 292. “33.9 in their research”. Cornelson et al.’s study measured Young's modulus at four distinct positions, with reported values ranging from 28.0 ± 12.44 kPa to 63.9 ± 27.37 kPa. For a more meaningful comparison, the authors should consider contrasting their results with those from Cornelson et al. obtained at the corresponding measurement position, rather than the numerically closest value.

Response: Thank you for your comment. We have improved this paragraph.

Lines 292 – 294. It is unclear how the results of Zardi et al. (2020) support the findings described in the previous sentence, as their reported Young’s modulus values ranged from 12.78 ± 3.59 kPa to 15.63 ± 6.44 kPa. These values fall outside the range measured by Cornelson et al. (2022) and the present study.

Response: Response: Thank you for your comment. We have improved this paragraph.

Lines 296 – 298. The authors are kindly requested to verify and correctly cite the appropriate studies for this sentence, using the citation format required by the jornal.

Response: corrected

Line 322. Kindly add appropriate references to support the authors’ claim.

Response: corrected

Lines 345 – 347. I apologize, but I was unable to identify any results in the text specifically referring to the dominant and non-dominant legs. I only found results related to the left and right legs. Therefore, I could not associate the findings with the statement in question.

Response: corrected in comments above.

  1. Practical Applications

Page 11, lines 425 – 426. As currently presented, this information is not accurate, since the results were not separated based on the dominant limb.

Response: corrected above.

Lines 432 – 434. I believe that, with some adjustments, this sentence would be more appropriate for the subsection “Limitations and Future Research”.

Response: corrected

  1. Conclusions

Page 12, line 440. Could the authors please add a space in the term “bySWE”, making it “by SWE”?

Response: Corrected.

  1. References

Page 12, lines 458 – 463. References 3 and 4 appear to be identical. Kindly review and update them accordingly. Line 526. The study by Mayorga-Vega et al. (2014) is included in the reference list but is not cited within the manuscript.  Page 14, lines 532 – 538. References 29 and 30 appear to be identical. Kindly review and update them accordingly. Lines 555 – 558. References 37 and 38 appear to be identical. Kindly review and update them accordingly.

Response: Thank your four your comments. We have improved this section.

Reviewer 3 Report

Comments and Suggestions for Authors

This manuscript aims to explore whether differences exist in sciatic nerve stiffness and SWV, assessed via SWE, and posterior chain mobility, measured through the AKE and ASLR tests, between dominant and non-dominant limbs, as well as between individuals with limited and adequate hamstring flexibility in healthy individuals. The authors have recruited a sufficient number of participants and obtained preliminary observational findings through data collection and analysis. However, several major issues need to be addressed:

  1. Manuscript Writing and Structure
    • The reference format throughout the manuscript must conform to the journal’s requirements. Specifically, in-text reference numbers should be placed in square brackets [ ] and positioned before the punctuation.
    • Some parts of the manuscript lack coherent logical flow. For example, the paragraph beginning at Line 40 should not logically connect to the paragraph starting at Line 49, but rather to the one beginning at Line 57. Similar issues of paragraph structure and coherence should be reviewed and corrected throughout the text.
    • The paragraph from lines 71 to 75 lacks logical coherence between the two sentences.
    • In the abstract, the full term for SWV is not provided, while in the table footnotes, the abbreviation is explained redundantly.
    • Additionally, Table 4 is missing the bottom border line.
    • The space was missing in Line 440 "... by SWE".
  2. Data Analysis Methodology
    • Both SWE values (in kPa) and SWV (in m/s) were used in the analysis. These two parameters are essentially proportional and can be converted into one another if tissue density is known. Therefore, the manuscript should justify the rationale for analyzing both quantities separately and examining their correlation. A strong correlation between these two measures is inherently expected and thus adds limited value without proper justification.
    • Similarly, the results of AKE and ASLR tests are inherently correlated, particularly when they are largely dependent on hamstring flexibility. This makes the findings presented in Table 4 appear rather trivial.
  3. Interpretation of Results
    • In the Discussion section, the manuscript states: “Although no direct correlation was observed between neural stiffness and flexibility, the increased SWE stiffness in individuals with reduced hamstring flexibility (p = 0.05) suggests that interventions aimed at improving neural mobility could positively influence range of motion.”
    • However, in the Conclusion, it states: “Individuals with greater limitation in posterior chain range of motion also demonstrated increased sciatic nerve stiffness, as measured by SWE.”
    • Based on the data presented in this study, there is no solid evidence to support such a conclusion. The authors should consider revising these statements to better reflect the actual findings.
  4. Data Visualization and Interpretation
    • The approach to data presentation is not intuitive. Providing only mean values and standard deviations does not allow readers to clearly understand the relationship between variables.
    • It is strongly recommended that the authors include graphical plots to illustrate trends in the data. For example, how do SWE and AKE or ASLR vary across the two groups (limited vs. adequate hamstring flexibility)? Visualizing these trends would greatly enhance the clarity and impact of the findings.

Author Response

Reviewer 3

This manuscript aims to explore whether differences exist in sciatic nerve stiffness and SWV, assessed via SWE, and posterior chain mobility, measured through the AKE and ASLR tests, between dominant and non-dominant limbs, as well as between individuals with limited and adequate hamstring flexibility in healthy individuals. The authors have recruited a sufficient number of participants and obtained preliminary observational findings through data collection and analysis. However, several major issues need to be addressed:

  1. Manuscript Writing and Structure

The reference format throughout the manuscript must conform to the journal’s requirements. Specifically, in-text reference numbers should be placed in square brackets [ ] and positioned before the punctuation.

Response: Thank you for your comment. We have corrected it.

Some parts of the manuscript lack coherent logical flow. For example, the paragraph beginning at Line 40 should not logically connect to the paragraph starting at Line 49, but rather to the one beginning at Line 57. Similar issues of paragraph structure and coherence should be reviewed and corrected throughout the text.

Response: Thank you for your comment. We have improved the clarity of the manuscript.

The paragraph from lines 71 to 75 lacks logical coherence between the two sentences.

Response: Thank you for your suggestion. We have improved this paragraph.

In the abstract, the full term for SWV is not provided, while in the table footnotes, the abbreviation is explained redundantly.

Response: We have corrected

Additionally, Table 4 is missing the bottom border line.

Response: corrected

The space was missing in Line 440 "... by SWE".

Response: corrected

  1. Data Analysis Methodology

Both SWE values (in kPa) and SWV (in m/s) were used in the analysis. These two parameters are essentially proportional and can be converted into one another if tissue density is known. Therefore, the manuscript should justify the rationale for analyzing both quantities separately and examining their correlation. A strong correlation between these two measures is inherently expected and thus adds limited value without proper justification.

Response: Thank you for this insightful comment. We agree that SWE values in kilopascals and shear wave velocity (SWV) in meters per second are mathematically related through the equation E = 3ρ, assuming constant tissue density. However, we chose to report and analyze both parameters for two main reasons: (1) to maintain consistency with prior literature, where both units are commonly presented, facilitating comparison; and (2) because not all elastography platforms automatically apply the same assumptions regarding tissue density, which may affect the derived stiffness values in kPa. We have clarified this rationale in the revised manuscript and acknowledge that the strong correlation observed between SWE and SWV was expected.

Similarly, the results of AKE and ASLR tests are inherently correlated, particularly when they are largely dependent on hamstring flexibility. This makes the findings presented in Table 4 appear rather trivial.

Response: Thank you for your observation. We agree that a strong correlation between AKE and ASLR is expected, as both tests assess posterior chain flexibility and are particularly influenced by hamstring extensibility. Our intention in including this correlation was to confirm the consistency between both measures and to provide internal validation of the functional assessments used in our study. We have now clarified this rationale in the manuscript.

  1. Interpretation of Results

In the Discussion section, the manuscript states: “Although no direct correlation was observed between neural stiffness and flexibility, the increased SWE stiffness in individuals with reduced hamstring flexibility (p = 0.05) suggests that interventions aimed at improving neural mobility could positively influence range of motion.” However, in the Conclusion, it states: “Individuals with greater limitation in posterior chain range of motion also demonstrated increased sciatic nerve stiffness, as measured by SWE.” Based on the data presented in this study, there is no solid evidence to support such a conclusion. The authors should consider revising these statements to better reflect the actual findings.

Response: Thank you for your suggestion. We have improved the discussion and conclusions section according to your valuable comments.

  1. Data Visualization and Interpretation

The approach to data presentation is not intuitive. Providing only mean values and standard deviations does not allow readers to clearly understand the relationship between variables.

It is strongly recommended that the authors include graphical plots to illustrate trends in the data. For example, how do SWE and AKE or ASLR vary across the two groups (limited vs. adequate hamstring flexibility)? Visualizing these trends would greatly enhance the clarity and impact of the findings.

Response: Thank you for your comment. We have added a figure (Figure 3) to illustrate the correlation between flexibility and SWE parameters.

Round 2

Reviewer 1 Report

Comments and Suggestions for Authors

Now it's suitable for publishing

Reviewer 2 Report

Comments and Suggestions for Authors

COMMENTS

     The authors have effectively addressed the suggestions and comments from the previous review, resulting in a significant improvement in the manuscript’s quality. However, to ensure it meets the journal’s standards, further revisions are required before it can be accepted for publication.

REITERATION OF PREVIOUS REVIEW SUGGESTIONS

     The following list shows my comments for individual sections:

A. Introduction

Line 51. “velocity”. The term “speed” is used to describe a scalar quantity, whereas “velocity” is employed for a velocity vector. So, I recommend that the authors replace “velocity” with “speed” throughout the manuscript for clarity and consistency.

Response: Thank you for this observation. While we acknowledge the distinction between “speed” and “velocity” in physics, we have chosen to retain the term “shear wave velocity” as it is the standardized terminology used throughout the scientific literature in elastography. The term “velocity” is consistently used in several articles and in the vast majority of peer-reviewed studies on shear wave elastography. Therefore, we believe maintaining the term ensures scientific accuracy and consistency with established conventions.

     I would like to suggest that authors consider performing a brief search on the PubMed database using the terms “shear wave velocity” and “hear wave speed”. This simple check reveals a notable difference in the number of publications associated with each term; over the last ten years, there are approximately 929 and 512 papers, respectively. This discrepancy largely reflects the sometimes imprecise and interchangeable use of these concepts, which can lead to confusion, particularly among non-specialist readers.

As someone working in the field of ultrasonic elastography and having collaborated as a reviewer for several indexed international journals, including Physics in Medicine and Biology, Journal of Bodywork & Movement Therapies, Measurement, Medical & Biological Engineering & Computing, Medical Engineering & Physics, and Frontiers in Physiology, I have often encouraged the appropriate and technically correct use of these terms. My aim is to contribute to greater clarity and standardization within the field.

     Furthermore, I would like to point out the common, yet potentially misleading, practice of using the term “shear modulus” as an indiscriminate synonym for “Young's modulus”. These are distinct quantities, with Young’s modulus being approximately three times greater than the shear modulus in isotropic materials. A clear distinction between these parameters is essential for the correct interpretation of results. Please note that the majority of commercial elastography equipment reports Young's modulus, not shear modulus.

Line 60. “(R. F. Ellis et al., 2012; Greening & Dilley, 2017)”. Please ensure that all study citations are formatted consistently, in accordance with the journal's guidelines. Furthermore, I suggest that the authors replace the reference to “Ellis et al. (2012)” with a more appropriate source, as the statement in question cannot be accurately attributed to these authors. While they utilized ultrasound imaging, they did not analyze the stiffness of the tissue under investigation.

Response: Thank you for your appreciation. We have corrected it.

    Thank you for your attention to the formatting of the citations in your manuscript. However, I would like to revisit a point I raised earlier concerning the citation of Ellias's study. Could you please elaborate on the reasoning that led you to consider this particular reference relevant to the specific context in which it was inserted within this work?

    I understand that the selection of references is a fundamental aspect of the argumentative construction of a study, and a clear justification for the inclusion of each citation significantly contributes to the robustness and overall comprehension of the work.

B. Methods

Lines 215 – 218. It is important to note that comparing the results from the right and left legs may not be the most appropriate approach for this type of analysis. Ideally, comparisons should be made between the dominant and nondominant limbs, as limb dominance can significantly influence neuromuscular, functional, and structural characteristics of the lower extremities.

Several studies have shown that the dominant limb typically exhibits greater strength, motor control, and coordination than the non-dominant limb (Hoffman et al., 1998; Sadeghi et al., 2000; Van Melick et al., 2017). Therefore, comparing the right and left sides without accounting for dominance may introduce bias and hinder accurate interpretation of the data.

Response: Thank you for pointing this out. It was a mistake on the tables. We have corrected it.

     I would like to respectfully suggest reconsidering the suggestion made in the previous review, as I continue to believe it is relevant for improving the manuscript. The observations detailed in the subsequent item also pertain to this point.

Lines 222 – 223. In the provided text, it is not clearly stated which specific parameters are measured in the Active Knee Extension Test and the Active Straight Leg Raise Test. The tests are mentioned only as variables in a correlation analysis, without further detail. Additionally, the text does not clarify whether the correlation was analyzed for the left, right, dominant, nondominant, or shortened leg. Providing this information would enhance the clarity and interpretability of the analysis.

Response: Thank you for your helpful comment. We have clarified that the correlation analysis was performed using values from all included limbs (both right and left legs), and that the parameters analyzed were the goniometric angle values obtained during the AKE and ASLR tests. This clarification has been added to the revised manuscript to improve clarity.

     “all included limbs (both left and right legs)”. I understand that the objective of the study focuses on the evaluation of the lower limbs. However, given the potential for functional or structural differences between the dominant and non-dominant limbs, it would be helpful to clarify whether this distinction was considered during data analysis. Many studies in the fields of biomechanics and exercise physiology explore such differences, and addressing this aspect could enhance the interpretation of the results.

    Additionally, to improve the reader’s understanding of the statistical analysis performed, it would be beneficial to specify which comparisons were made between the evaluated groups or variables.

  1. Results

Page 6, Table 1. If the total number of right legs analyzed is 25 (25 participants), could you please clarify how there are 43 instances of right limb dominance reported in Table 1?

Response: Thank you for your comment. 43, is the number of right limbs.

     Considering the number of participants (25) and the total number of lower limbs analyzed (49), there appears to be an inconsistency in the reported number of right-dominant legs (n = 43).

With a sample of 25 participants, the maximum number of right-dominant legs would be 25, assuming all individuals were right-leg dominant. Therefore, the reported value of n = 43 seems incompatible with the sample size.

     I kindly request that the authors review and clarify how the value of n = 43 for right-dominant legs was determined, given the total number of participants and limbs analyzed. A possible explanation could be a typographical error or an alternative method of data classification that is not clearly described in the text.

Table 2. Kindly ensure that the degree unit is represented by the “ º ” symbol.

Response: corrected.

     I suggest double-checking the aforementioned section. While it was indicated that a correction was made, the change does not appear to be present in the current text.

Table 4. For the sake of clarity, it is recommended that the authors describe the specific parameters rather than simply naming the test. For instance, SWE is commonly used to refer to the Shear Wave Elastography technique, which can be employed to measure either shear wave speed and/or Young’s modulus.

Response: Thank you for your appreciation. We have improved it.

I suggest double-checking the aforementioned section. While it was indicated that a correction was made, the change does not appear to be present in the current text.

Kindly ensure that the degree unit is represented by the “ º ” symbol.

Could you please clarify if the data for the non-dominant leg is shown in the second or third column of this table?

D. Discussion

Lines 354 – 355. Kindly add appropriate references to support the authors’ claim.

Response: corrected.

    I suggest double-checking the aforementioned section. While it was indicated that a correction was made, the change does not appear to be present in the current text.

NEW COMMENTS

The following observations and recommendations are based on the analysis of the revised version of the manuscript.

A. Introduction

Page 2, lines 80 – 81. Could you please add appropriate references to support the claim made in this sentence?

B. Methods

Page 3, lines 126 – 127. I would like to kindly reiterate an observation made in the previous review regarding the unit of measurement for angles. The correct symbol to use is “º” (degree), rather than the unit currently used in the manuscript.

Page 5, lines 178 – 182. I have a question regarding the cited excerpt. Did the authors use the shear wave speed to estimate the shear modulus (μ)? If so, I believe the equation presented (“E = 3ρv²”) may not be appropriate for this purpose. Additionally, it would be helpful to clarify this point in the text, as ultrasound equipment typically provides both shear wave velocity and Young’s modulus directly.

In an ideal elastic and isotropic medium, the propagation speed of shear waves (cs​) is mathematically linked to the shear modulus (μ) through the relationship μ=ρcs2​, where ρ represents the density of the biological tissue, typically assumed to be 1,000 kg/m³. In the case of nearly incompressible materials, such as most biological tissues, the shear modulus approximates one-third of the Young’s modulus (μ ≈ E/3). Consequently, under the assumption of pure elasticity, the Young’s modulus can be estimated using the formula E = 3ρcs2, indicating that increases in shear wave speed correspond to increases in tissue stiffness (Young Modulus).

Page 6, lines 224 – 225. While this information is helpful for describing the data used, I suggest that the description of the statistical analysis be expanded to more clearly explain how each test was applied in relation to the study’s objectives. Providing the rationale behind each statistical test and how it aligns with the overall goals of the research would enhance the reader’s understanding of the methodology and support a more accurate interpretation of the results.

C. Results

Table 3. Kindly ensure that the degree unit is represented by the “ º ” symbol.

Figure 3. I recommend that Figure 3 be discussed in greater depth within the main text, incorporating some of the valuable details already provided in the figure caption. This additional discussion would enhance the reader’s understanding of the results presented. I also note that the degree unit is used correctly in this figure.

D. Discussion

Page 10, lines 348 and 350. Could you please ensure that the degree unit is represented by the “º” symbol?

Line 330. I would like to kindly request the inclusion of the unit of measurement corresponding to the presented Young's modulus values ('12.78 – 15.63').

Additionally, I suggest inserting a space before and after the en dash “–“ (used here to indicate a range of values), for improved clarity in the presentation."

Line 350.  Once again, I would like to respectfully request that the authors use the correct unit symbol for degrees, which is “º”.

Author Response

Dear Reviewer,

Thank you very much for your comments aimed at improving the quality of the manuscript. We greatly appreciate your excellent insights based on your experience in the field of sonoelastography research. We have attempted to address all the issues you raised and are displaying the response comments as "R2 Response." We hope you find the adaptations appropriate.

Sincerely,

The Authors

 The authors have effectively addressed the suggestions and comments from the previous review, resulting in a significant improvement in the manuscript’s quality. However, to ensure it meets the journal’s standards, further revisions are required before it can be accepted for publication.

REITERATION OF PREVIOUS REVIEW SUGGESTIONS

     The following list shows my comments for individual sections:

  1. Introduction

Line 51. “velocity”. The term “speed” is used to describe a scalar quantity, whereas “velocity” is employed for a velocity vector. So, I recommend that the authors replace “velocity” with “speed” throughout the manuscript for clarity and consistency.

Response: Thank you for this observation. While we acknowledge the distinction between “speed” and “velocity” in physics, we have chosen to retain the term “shear wave velocity” as it is the standardized terminology used throughout the scientific literature in elastography. The term “velocity” is consistently used in several articles and in the vast majority of peer-reviewed studies on shear wave elastography. Therefore, we believe maintaining the term ensures scientific accuracy and consistency with established conventions.

     I would like to suggest that authors consider performing a brief search on the PubMed database using the terms “shear wave velocity” and “hear wave speed”. This simple check reveals a notable difference in the number of publications associated with each term; over the last ten years, there are approximately 929 and 512 papers, respectively. This discrepancy largely reflects the sometimes imprecise and interchangeable use of these concepts, which can lead to confusion, particularly among non-specialist readers.

As someone working in the field of ultrasonic elastography and having collaborated as a reviewer for several indexed international journals, including Physics in Medicine and Biology, Journal of Bodywork & Movement Therapies, Measurement, Medical & Biological Engineering & Computing, Medical Engineering & Physics, and Frontiers in Physiology, I have often encouraged the appropriate and technically correct use of these terms. My aim is to contribute to greater clarity and standardization within the field.

     Furthermore, I would like to point out the common, yet potentially misleading, practice of using the term “shear modulus” as an indiscriminate synonym for “Young's modulus”. These are distinct quantities, with Young’s modulus being approximately three times greater than the shear modulus in isotropic materials. A clear distinction between these parameters is essential for the correct interpretation of results. Please note that the majority of commercial elastography equipment reports Young's modulus, not shear modulus.

Response R2: Thank you very much for your valuable comment and for sharing your expertise in the field of ultrasonic elastography. We appreciate your detailed explanation and agree that using technically precise terminology enhances clarity. Therefore, we have revised the manuscript and replaced the term “velocity” with “speed” throughout the text to reflect the scalar nature of this measurement and to ensure consistency with the correct physical definition.

We also thank you for the clarification regarding the distinction between Young’s modulus and shear modulus. We have reviewed our manuscript carefully to ensure that no misuse or confusion between these terms remains.

Line 60. “(R. F. Ellis et al., 2012; Greening & Dilley, 2017)”. Please ensure that all study citations are formatted consistently, in accordance with the journal's guidelines. Furthermore, I suggest that the authors replace the reference to “Ellis et al. (2012)” with a more appropriate source, as the statement in question cannot be accurately attributed to these authors. While they utilized ultrasound imaging, they did not analyze the stiffness of the tissue under investigation.

Response: Thank you for your appreciation. We have corrected it.

    Thank you for your attention to the formatting of the citations in your manuscript. However, I would like to revisit a point I raised earlier concerning the citation of Ellias's study. Could you please elaborate on the reasoning that led you to consider this particular reference relevant to the specific context in which it was inserted within this work?

    I understand that the selection of references is a fundamental aspect of the argumentative construction of a study, and a clear justification for the inclusion of each citation significantly contributes to the robustness and overall comprehension of the work.

Response R2: Thank you for your detailed observation. We sincerely apologize for the misattribution. The inclusion of “Ellis et al. (2012)” in this context was the result of an error in our reference manager software. Our intention was to cite the systematic review by Ciuffreda et al. 2024.

  1. Methods

Lines 215 – 218. It is important to note that comparing the results from the right and left legs may not be the most appropriate approach for this type of analysis. Ideally, comparisons should be made between the dominant and nondominant limbs, as limb dominance can significantly influence neuromuscular, functional, and structural characteristics of the lower extremities.

Several studies have shown that the dominant limb typically exhibits greater strength, motor control, and coordination than the non-dominant limb (Hoffman et al., 1998; Sadeghi et al., 2000; Van Melick et al., 2017). Therefore, comparing the right and left sides without accounting for dominance may introduce bias and hinder accurate interpretation of the data.

Response: Thank you for pointing this out. It was a mistake on the tables. We have corrected it.

     I would like to respectfully suggest reconsidering the suggestion made in the previous review, as I continue to believe it is relevant for improving the manuscript. The observations detailed in the subsequent item also pertain to this point.

Response R2: Thank you very much for revisiting this important methodological point. We would like to confirm that we fully agreed with your previous suggestion and already implemented the correction in the revised version of the manuscript. The analysis now clearly compares dominant versus non-dominant limbs, both in the text and in the tables, and we have ensured consistency throughout the results and discussion sections.

Lines 222 – 223. In the provided text, it is not clearly stated which specific parameters are measured in the Active Knee Extension Test and the Active Straight Leg Raise Test. The tests are mentioned only as variables in a correlation analysis, without further detail. Additionally, the text does not clarify whether the correlation was analyzed for the left, right, dominant, nondominant, or shortened leg. Providing this information would enhance the clarity and interpretability of the analysis.

Response: Thank you for your helpful comment. We have clarified that the correlation analysis was performed using values from all included limbs (both right and left legs), and that the parameters analyzed were the goniometric angle values obtained during the AKE and ASLR tests. This clarification has been added to the revised manuscript to improve clarity.

     “all included limbs (both left and right legs)”. I understand that the objective of the study focuses on the evaluation of the lower limbs. However, given the potential for functional or structural differences between the dominant and non-dominant limbs, it would be helpful to clarify whether this distinction was considered during data analysis. Many studies in the fields of biomechanics and exercise physiology explore such differences, and addressing this aspect could enhance the interpretation of the results.

    Additionally, to improve the reader’s understanding of the statistical analysis performed, it would be beneficial to specify which comparisons were made between the evaluated groups or variables.

Response R2: Thank you for your detailed and constructive comment. We have carefully revised the manuscript to improve the clarity of the correlation analysis and the variables involved. Specifically, we now state that the correlation analysis was performed using the goniometric angle values obtained from the Active Knee Extension (AKE) and Active Straight Leg Raise (ASLR) tests, which were applied to all included limbs.

We have also explicitly mentioned that limb dominance was considered in the between-group comparisons (dominant vs. non-dominant), as suggested in your previous comments, and this distinction was applied to both descriptive and inferential analyses.

Additionally, we now clarify that the correlation matrix did not group limbs by dominance or flexibility, but analyzed the total sample of measured limbs, as the focus was to explore general associations between functional flexibility measures and elastographic parameters in healthy participants.

  1. Results

Page 6, Table 1. If the total number of right legs analyzed is 25 (25 participants), could you please clarify how there are 43 instances of right limb dominance reported in Table 1?

Response: Thank you for your comment. 43, is the number of right limbs.

     Considering the number of participants (25) and the total number of lower limbs analyzed (49), there appears to be an inconsistency in the reported number of right-dominant legs (n = 43).

With a sample of 25 participants, the maximum number of right-dominant legs would be 25, assuming all individuals were right-leg dominant. Therefore, the reported value of n = 43 seems incompatible with the sample size.

     I kindly request that the authors review and clarify how the value of n = 43 for right-dominant legs was determined, given the total number of participants and limbs analyzed. A possible explanation could be a typographical error or an alternative method of data classification that is not clearly described in the text.

Response R2: Thank you for this careful observation. You are absolutely correct — the value of n = 43 for right-dominant limbs was the result of a data extraction error during the statistical analysis. The correct number of participants with right lower-limb dominance is 22 out of 25, as confirmed after reviewing the original dataset. We have now corrected Table 1 accordingly and ensured consistency throughout the manuscript. We appreciate your attention to this detail, which helped us to improve the accuracy of our data presentation.

Table 2. Kindly ensure that the degree unit is represented by the “ º ” symbol.

Response: corrected.

     I suggest double-checking the aforementioned section. While it was indicated that a correction was made, the change does not appear to be present in the current text.

Table 4. For the sake of clarity, it is recommended that the authors describe the specific parameters rather than simply naming the test. For instance, SWE is commonly used to refer to the Shear Wave Elastography technique, which can be employed to measure either shear wave speed and/or Young’s modulus.

Response: Thank you for your appreciation. We have improved it.

I suggest double-checking the aforementioned section. While it was indicated that a correction was made, the change does not appear to be present in the current text.

Kindly ensure that the degree unit is represented by the “ º ” symbol.

Could you please clarify if the data for the non-dominant leg is shown in the second or third column of this table?

Response R2: Thank you for your comment. We have corrected it.

  1. Discussion

Lines 354 – 355. Kindly add appropriate references to support the authors’ claim.

Response: corrected.

    I suggest double-checking the aforementioned section. While it was indicated that a correction was made, the change does not appear to be present in the current text.

Response R2: Thank you for your comment. We have changed the reference.

NEW COMMENTS

The following observations and recommendations are based on the analysis of the revised version of the manuscript.

  1. Introduction

Page 2, lines 80 – 81. Could you please add appropriate references to support the claim made in this sentence?

Response: Thank you for your comment. We have added the reference.

  1. Methods

Page 3, lines 126 – 127. I would like to kindly reiterate an observation made in the previous review regarding the unit of measurement for angles. The correct symbol to use is “º” (degree), rather than the unit currently used in the manuscript.

Response: Thank you for your comment. We have changed the symbol.

Page 5, lines 178 – 182. I have a question regarding the cited excerpt. Did the authors use the shear wave speed to estimate the shear modulus (μ)? If so, I believe the equation presented (“E = 3ρv²”) may not be appropriate for this purpose. Additionally, it would be helpful to clarify this point in the text, as ultrasound equipment typically provides both shear wave velocity and Young’s modulus directly.

In an ideal elastic and isotropic medium, the propagation speed of shear waves (cs​) is mathematically linked to the shear modulus (μ) through the relationship μ=ρcs2​, where ρ represents the density of the biological tissue, typically assumed to be 1,000 kg/m³. In the case of nearly incompressible materials, such as most biological tissues, the shear modulus approximates one-third of the Young’s modulus (μ ≈ E/3). Consequently, under the assumption of pure elasticity, the Young’s modulus can be estimated using the formula E = 3ρcs2, indicating that increases in shear wave speed correspond to increases in tissue stiffness (Young Modulus).

Response: Thank you for your appreciation. We have corrected in the main text. To address this, we have revised the manuscript to clarify that no manual calculations were performed, and that reported values correspond directly to the system outputs. We have also removed any ambiguous statements regarding formula derivation to avoid confusion.
We are grateful for your guidance, which helped us improve the precision and clarity of our methodological description.

Page 6, lines 224 – 225. While this information is helpful for describing the data used, I suggest that the description of the statistical analysis be expanded to more clearly explain how each test was applied in relation to the study’s objectives. Providing the rationale behind each statistical test and how it aligns with the overall goals of the research would enhance the reader’s understanding of the methodology and support a more accurate interpretation of the results.

Response: Thank you for your comment. We have reformulated this part of the statistical analysis description.

  1. Results

Table 3. Kindly ensure that the degree unit is represented by the “ º ” symbol.

Response: Corrected.

Figure 3. I recommend that Figure 3 be discussed in greater depth within the main text, incorporating some of the valuable details already provided in the figure caption. This additional discussion would enhance the reader’s understanding of the results presented. I also note that the degree unit is used correctly in this figure.

Response: thank you for your suggestion. We have added a brief description of the figure in the main text.

  1. Discussion

Page 10, lines 348 and 350. Could you please ensure that the degree unit is represented by the “º” symbol?

Response: corrected.

Line 330. I would like to kindly request the inclusion of the unit of measurement corresponding to the presented Young's modulus values ('12.78 – 15.63').

Additionally, I suggest inserting a space before and after the en dash “–“ (used here to indicate a range of values), for improved clarity in the presentation."

Response: corrected.

Line 350.  Once again, I would like to respectfully request that the authors use the correct unit symbol for degrees, which is “º”.

Response: corrected

Reviewer 3 Report

Comments and Suggestions for Authors

Thanks for addressing the comments!

Author Response

Thank you for your attention.

Sincerely,
The Authors